# Exploring SGLT-2 Inhibitors: Benefits beyond the Glucose-Lowering Effect—What Is New in 2023?

Clipper F. Young [1,*], Neeka Farnoudi [2], Jenny Chen [3] and Jay H. Shubrook [1]

[1] Department of Clinical Sciences and Community Health, Touro University California College of Osteopathic Medicine, Vallejo, CA 94592, USA; jshubroo@touro.edu
[2] Loma Linda University Health, Loma Linda, CA 92354, USA; nfarnoud@student.touro.edu
[3] Kaiser Permanente, San Rafael, CA 94903, USA; jchen30@student.touro.edu
[*] Correspondence: cyoung6@touro.edu

**Abstract:** Sodium-glucose cotransporter-2 (SGLT-2) inhibitors were once known as a class of glycemic-lowering agents to treat type 2 diabetes. As the evolving evidence from recent cardiorenal trials on these agents has shown—e.g., EMPA-REG OUTCOME, DECLARE-TIMI 58, CANVAS Program, DAPA-CKD—disclosing their benefits beyond glycemic management, SGLT-2 inhibitors have stimulated a shift in the management of T2DM and its comorbidities, specifically preventing cardiovascular events in people with ASCVD, preventing heart failure hospitalizations, and delaying the progression of chronic kidney disease. As a result, their usage beyond glycemic management has been included in clinical practice guidelines. Although SGLT-2 inhibitors have shown promising results in cardiorenal outcomes, patients have not had equal access to these agents, at least in the United States, suggesting a systemic issue of health inequity. This review article explores the mechanisms by which cardiorenal benefits are offered, the results of the landmark clinical trials for these agents, and their place in therapy.

**Keywords:** SGLT-2 inhibitors; renoprotective benefits; cardiovascular benefits; heart failure benefits; pharmacoequity





## 1. Introduction

The successful management of type 2 diabetes mellitus (T2DM) requires timely multifaceted interventions, which may include pharmacotherapy, lifestyle modifications (e.g., physical activity and a modified diet), psychosocial care, glucose self-monitoring, and diabetes self-management education and support (DSMES). The overall strategies are: (1) to minimize the risks of complications resulting from diabetes or to manage co-existing chronic conditions (e.g., chronic kidney disease, cardiovascular disease, heart failure); (2) to achieve glycemic management and weight reduction goals.

The World Health Organization (WHO) has reported that 422 million people have been impacted by diabetes globally, especially those in low- and middle-income countries [1]. According to the 2022 National Diabetes Statistics Report published by the Centers for Disease Control and Prevention (CDC), 37.3 million people in the United States (U.S.) were living with diabetes in 2019, accounting for those who were diagnosed and undiagnosed [2]. This was equivalent to 11.3% of the US population having T2DM, accounting for nearly 90% to 95% of all diabetes cases [2,3]. The estimated total cost, direct and indirect, for managing patients with diabetes in the US was USD 327 billion in 2017 [4]. Annually, the American Diabetes Association (ADA) publishes and revises their guidelines for managing diabetes—*Standards of Care in Diabetes (SOC)*—providing the most up-to-date recommendations based on the newest evidence in the field [5].

Chronic suboptimal glycemic management in diabetes has been well-documented, leading to all forms of complications, resulting in preventable macrovascular complications (e.g., stroke and myocardial infarction), microvascular complications (e.g., nephropathy,

retinopathy, neuropathy), and other related health complications (e.g., amputation of toes, feet, or lower limbs). Furthermore, the mortality rate of cardiovascular (CV) disease in people with T2DM is two to four times higher compared to individuals without T2DM [6].

According to the 2022 CDC National Diabetes Statistics Report, diabetes was the leading cause of end-stage kidney disease (ESKD) in that year. Among adults 18 years or older with ESKD, 39.2% of the cases were secondary to diabetes [7]. In adults with diabetes, 1.8 million hospitalizations were due to major CV diseases (74.4 per 1000 adults with T2DM), including 440,000 for ischemic heart disease and 334,000 for stroke [7]. In addition, the prevalence of heart failure (HF) in people with T2DM is between 9% and 22%, which is four times higher than the general population [8]. In 2019, the WHO reported that an estimated 1.5 million deaths were directly caused by diabetes, and diabetes was ranked as the ninth leading cause of death worldwide [1]. Among these 1.5 million deaths relating to diabetes, 48% occurred before the age of 70 years old [1].

To minimize the associated mortality and improve the management strategies for T2DM, therapeutic advances have been implemented to support the glycemic management of the condition and reduce the risks for diabetes-related complications. Sodium-glucose cotransporter-2 inhibitors (SGLT-2i) are the newest class of agents to be added to the plethora of existing pharmacotherapeutic options for T2DM. SGLT2i, additionally, are supported with evidence-based benefits in preventing or delaying the negative health outcomes of T2DM, such as HF, chronic kidney disease (CKD), and atherosclerotic cardiovascular disease (ASCVD). The 2023 ADA SOC guidelines recommend that SGLT-2i be added to medication regimens for T2DM management independent of the individuals' baseline A1c levels, their individualized A1c targets, or usage of metformin if they have a previously mentioned co-morbid condition [5]. They are the preferred treatment in people with heart failure and CKD with albuminuria, regardless of HbA1c, and are offered as one of two preferred agents in people with ACDVD and CKD without albuminuria [5]. Due to the positive clinical outcomes resulting from the major clinical trials reporting these agents' non-glycemic benefits—CKD, ASCVD, and HF—in patients with or without T2DM, this article focuses on reviewing the mechanisms through which these agents provide their non-glycemic benefits and the existing evidence, as well as exploring how the evidence has shaped and enhanced clinical practice.

The focus of this review paper will cover how SGLT-2i can provide organ protection through supported evidence in landmark trials and plausible theories. Reviewing each disease state will start with the normal physiology and diabetes-related pathophysiology. Most importantly, this article will discuss how SGLT-2i can be incorporated into clinical practice for those living with T2DM.

## 1.1. An Overview of SGLT-2 Inhibitors

The SGLT-2i that have been approved in the U.S. include canagliflozin (2013), empagliflozin (2014), dapagliflozin (2014), ertugliflozin (2017), bexagliflozin (2023), and sotagliflozin (2023). Table 1 summarizes the glycemic effects and weight reduction in the SGLT-2 inhibitors approved before 2023. The newer SGLT-2 inhibitors have limited approvals, including bexagliflozin approved for glycemic management only in T2DM [9], and sotagliflozin approved for treating heart failure in patients with T2DM [10].

**Table 1.** Glycemic Effects and Weight Reduction in SGLT-2 Inhibitors.

| Efficacy: SGLT-2 Inhibitors as Monotherapy in Placebo-Controlled Studies | | | | | | | | |
|---|---|---|---|---|---|---|---|---|
| | Canagliflozin [11] Invokana 26-week study | | Dapagliflozin [12] Farxiga 24-week study | | Empagliflozin [13] Jardiance 24-week study | | Ertugliflozin [14] Steglatro 26-week study | |
| | 100 mg | 300 mg | 5 mg | 10 mg | 10 mg | 25 mg | 5 mg | 15 mg |
| A1C (%) Difference from Placebo | −0.91 | −1.16 | −0.5 | −0.7 | −0.7 | −0.9 | −0.6 | −0.7 |
| Fasting Blood Glucose (mg/dL) | −36 | −43 | −19.9 | −24.7 | −31 | −36 | −19.4 | −24.8 |
| 2-Hour Post Prandial Glucose (mg/dL) | −48 | −64 | --- | --- | --- | --- | --- | --- |
| | Canagliflozin 26-week study | | Dapagliflozin 24-week study | | Empagliflozin 24-week study | | Ertugliflozin 26-week study | |
| | 100 mg | 300 mg | 5 mg | 10 mg | 10 mg | 25 mg | 5 mg | 15 mg |
| Body Weight Reduction (kg) | −2.2 | −3.3 | --- | --- | −2.5 | −2.8 | −2 | −2.1 |
| Body Weight Reduction (lb) | −4.84 | −7.26 | --- | --- | −5.5 | −6.16 | −4.4 | −4.62 |

*1.2. Mechanism of Action*

SGLT-2i preserve nephron functionality by competitively binding to SGLT-2 cotransporters in a reversible manner to inhibit glucose and sodium reabsorption [15]. In normal physiology, nearly 100% of the glucose molecules that enter renal tubules (RT) are reabsorbed into the body unless the plasma glucose concentrations exceed 180 mg/dL [16]. This is primarily accomplished by sodium-glucose cotransporters (SGLT)—the two major isoforms being SGLT-1 and SGLT-2, the latter accounting for 90% of glucose reabsorption in the proximal convoluted tubules (PCT) [16]. SGLT-2i reduce the renal threshold for glucose from ~180 mg/dL to ~40 mg/dL, thus increasing the elimination of glucose in the blood through glucosuria [17]. Table 2 lists the FDA-approved indications, recommended dosing, and the possible combinations of SGLT2i with other oral glucose-lowering agents.

*1.3. Adverse Effects*

The adverse profile of these agents has become more favorable over time. All of the SGLT-2i have warnings about the increased risk of urinary tract infection (UTI) and genital mycotic infections [9–14]. These medications have also been implicated for Fournier's Gangrene [9–14]. Finally, euglycemic diabetic ketoacidosis has been seen with SGLT-2i and should not be used in people with type 1 diabetes [9–14]. Originally, there was a concern about canagliflozin and distal limb amputations, but this has not been seen in the post-marketing data. In 2020, the FDA removed canagliflozin's black-boxed warning (BBW) regarding the risk of leg and foot amputations [18]. At present, there are no BBWs associated with any of the FDA-approved SGLT-2i. The risk of developing hypoglycemia from SGLT-2i is low because glucose reabsorption in the kidneys is not insulin-regulated, and there is some reabsorption of glucose through the uninhibited SGLT-2 cotransporters [15,18]. Hypoglycemia is more likely to occur if an SGLT-2 inhibitor is combined with other glucose-lowering agents rather than as a monotherapy [19]. Table 3 summarizes the safety profiles of the SGLT-2 inhibitors approved before 2023.

**Table 2.** SGLT-2 Inhibitors Drug Label Information.

| | Canagliflozin [11] 100 mg, 300 mg | Dapagliflozin [12] 5 mg, 10 mg | Empagliflozin [13] 10 mg, 25 mg | Ertugliflozin [14] 5 mg, 15 mg |
|---|---|---|---|---|
| FDA-approved Indications | 1. Used "as an adjunct to diet and exercise to improve glycemic control in adults with type 2 diabetes mellitus" 2. Reducing "the risk of major adverse cardiovascular events in adults with type 2 diabetes mellitus and established cardiovascular disease" 3. Reducing "the risk of end-stage kidney disease, doubling of serum creatinine, cardiovascular death, and hospitalization for heart failure in adults with type 2 diabetes mellitus and diabetic nephropathy with albuminuria" | 1. Used "as an adjunct to diet and exercise to improve glycemic control in adults with type 2 diabetes mellitus" 2. Reducing "the risk of hospitalization for heart failure in adults with type 2 diabetes mellitus and either established cardiovascular disease or multiple cardiovascular risk factors" 3. Reducing "the risk of cardiovascular death and hospitalization for heart failure in adults with heart failure with reduced ejection (NYHA class II–IV)" 4. Reducing "the risk of sustained eGFR decline, end stage kidney disease cardiovascular death and hospitalization for heart failure in adults with chronic kidney disease at risk of progression" | 1. Used "as an adjunct to diet and exercise to improve glycemic control in adults with type 2 diabetes 2. Reducing "the risk of cardiovascular death and hospitalization for heart failure in adults with heart failure" 3. Reducing "the risk of cardiovascular death in adults with type 2 diabetes mellitus and established cardiovascular disease" | 1. Used "as an adjunct to diet and exercise to improve glycemic control in adults with type 2 diabetes |
| Dosing | Based on eGFR: • eGFR >60: 100 mg once daily. If tolerated, may increase to 300 mg once daily for additional glycemic control. • eGFR 30–60: 100 mg once daily • eGFR <30: "Initiation is not recommended, however patients with albuminuria greater than 300 mg/day may continue 100 mg once daily to reduce the risk of ESRD, doubling of SCr, CV death, and hospitalization for HF" • On dialysis: contraindicated | Based on eGFR: • eGFR >45: ○ Glycemic control only: Start with 5 mg once daily. If tolerated, may increase to 10 mg once daily for additional glycemic control. ○ Other indications: start with 10 mg once daily • eGFR 25–45: 10 mg once daily • eGFR <25: "Initiation is not recommended, however patients may continue 10 mg once daily to reduce risk of eGFR decline, ESRD, CV death, and hospitalization for HF" • On dialysis: contraindicated | • eGFR ≥30: ○ Glycemic control only: Start with 10 mg once daily. If tolerated, may increase to 25 mg once daily for additional glycemic control. • eGFR <30: ○ Patients with type 2 diabetes and established CV disease: no recommendations due to insufficient data • eGFR <20: ○ Patients with HF: no recommendations due to insufficient data • On dialysis: contraindicated | Based on eGFR: • eGFR ≥45: ○ Glycemic control only: Start with 5 mg once daily. If tolerated, may increase to 15 mg once daily for additional glycemic control. • eGFR <45: not recommended • On dialysis: contraindicated |
| Combination Therapies | canagliflozin and metformin | dapagliflozin and metformin; dapagliflozin and saxagliptin | empagliflozin and linagliptin; empagliflozin and metformin; empagliflozin, metformin and linagliptin | ertugliflozin and metformin; ertugliflozin and sitagliptin |

MACE: major adverse cardiovascular events (CV death, myocardial infarction, ischemic stroke); DKD: diabetic kidney disease; ESRD: end-stage renal disease; ESKD: end-stage kidney disease; SCr: serum creatinine; CV: cardiovascular; HHF: hospitalization for heart failure.

**Table 3.** Safety Profile of SGLT-2 Inhibitors.

| | Canagliflozin [11] 100 mg, 300 mg | Dapagliflozin [12] 5 mg, 10 mg | Empagliflozin [13] 10 mg, 25 mg | Ertugliflozin [14] 5 mg, 15 mg |
|---|---|---|---|---|
| Warnings/Precautions | <ul><li>Lower limb amputation</li><li>Volume depletion</li><li>Ketoacidosis (euglycemia or hyperglycemia)</li><li>Urosepsis and pyelonephritis</li><li>Hypoglycemia with concomitant use with insulin and insulin secretagogues</li><li>Necrotizing fasciitis of the perineum</li><li>Genital mycotic infections</li><li>Hypersensitivity reactions</li><li>Bone fractures</li></ul> | <ul><li>Volume depletion</li><li>Ketoacidosis (euglycemia or hyperglycemia)</li><li>Urosepsis and pyelonephritis</li><li>Hypoglycemia with concomitant use with insulin and insulin secretagogues</li><li>Necrotizing fasciitis of the perineum</li><li>Genital mycotic infections</li></ul> | <ul><li>Volume depletion</li><li>Ketoacidosis (euglycemia or hyperglycemia)</li><li>Urosepsis and pyelonephritis</li><li>Hypoglycemia with concomitant use with insulin and insulin secretagogues</li><li>Necrotizing fasciitis of the perineum</li><li>Genital mycotic infections</li><li>Hypersensitivity reactions</li></ul> | <ul><li>Lower limb amputation</li><li>Volume depletion</li><li>Ketoacidosis (euglycemia or hyperglycemia)</li><li>Urosepsis and pyelonephritis</li><li>Hypoglycemia with concomitant use with insulin and insulin secretagogues</li><li>Necrotizing fasciitis of the perineum</li><li>Genital mycotic infections</li></ul> |
| Adverse Events (incidence ≥ 5%) | Female genital mycotic infections; urinary tract infections; increased urination | Female genital mycotic infections; nasopharyngitis; urinary tract infections | Female genital mycotic infections; urinary tract infections | Female genital mycotic infections |
| Contraindications | Hypersensitivity to canagliflozin with serious reactions such as anaphylaxis or angioedema; patients on dialysis | Hypersensitivity to dapagliflozin with serious reactions such as anaphylaxis, urticaria, or angioedema; patients on dialysis | Hypersensitivity to empagliflozin with serious reactions such as urticaria or angioedema; patients on dialysis | Hypersensitivity to ertugliflozin with serious reactions such as angioedema; patients on dialysis |

## 2. Relevant Sections

### 2.1. Renoprotective Benefits

People with T2DM with suboptimally managed glucose levels are at an increased risk of microvascular complications, namely nephropathy, retinopathy, and peripheral neuropathy. Three landmark trials—e.g., CREDENCE, DAPA-CKD, and EMPA-KIDNEY—reported strong evidence of delaying the progression of diabetic kidney disease (DKD) or death from renal causes by including canagliflozin, dapagliflozin, or empagliflozin, respectively, into one's diabetes management plan. The risk factors for DKD include a long history of diabetes, hypertension, obesity, hyperlipidemia, etc. [20]. At present, canagliflozin and dapagliflozin are the only two SGLT-2i with FDA approval to delay disease progression independent of the glycemic state [5].

### 2.1.1. What Are the Proposed Theories or Potential Mechanisms of SGLT-2 Inhibitors in Providing Renal Benefits?

The mechanism of action for SGLT-2i is to inhibit glucose and sodium reabsorption by binding to SGLT-2 cotransporters [15]. The renal threshold for glucose is ~180 mg/dL, which allows glucose to be reabsorbed in the kidneys by the proximal convoluted tubules (PCT) [16]. SGLT2i lower the renal threshold for glucose reabsorption to ~40 mg/dL, which enhances glucosuria [13]. Within the first 48 to 72 h of administration, SGLT2i can remove ~100 mEq of sodium and ~400 to 700 mL of water [15,17]. This mechanism relies on the SGLT-2i entering into the tubules. These medications lose therapeutic effect at an eGFR of <45 mL/min. As discussed later in this paper, these medications will still have benefits in kidney disease and heart disease, even at lower eGFR.

As an SGLT-2i limits the reabsorption of sodium and glucose, osmotic diuresis occurs when water is pulled towards a higher concentration of sodium and glucose into the renal tubule. A transient drop in eGFR immediately following the initiation of SGLT-2i is seen and may be a result of diuresis and volume contraction [15,19]. Although SGLT2i modestly lower blood pressure from the fluid shifts, they should not be used as maintenance therapy for hypertension. This paper describes the proposed mechanisms of renal protection through the use of SGLT2i; osmotic diuresis was noted earlier, and the remaining includes tubuloglomerular feedback, tubular oxygenation, tubular energetics, the uricosuric effect, and reduced renal inflammation [15].

The upregulated expression of SGLT-2 cotransporters in people with T2DM increases the reabsorption of glucose and sodium in the PCT [19]. In people with diabetes, the kidneys upregulate the capacity of SGLT-2 co-transporters in the PCT to reabsorb ~500–600 g of glucose per day compared to 400 g of glucose per day in those who do not have diabetes [19]. With the introduction of an SGLT-2 inhibitor, glucose, and sodium reabsorption are decreased in the PCT, thus activating tubuloglomerular feedback (TGF) through the enhanced sodium delivery to the macula densa. Through the TGF mechanism, the macula densa cells increase afferent arteriole vasoconstriction and decrease efferent arteriole vasoconstriction [15]. Thus, SGLT-2i suppress hyperfiltration indirectly by reducing the intraglomerular pressure, leading to a temporary decrease in eGFR and albuminuria excretion [15,19]. The temporary decline in eGFR reduces the harmful intraglomerular hyperfiltration in DKD, which is considered a maladaptive compensatory state that precedes albuminuria or a progressive decline in renal function [19]. This, in turn, preserves renal function or delays the eGFR decline [19].

The PCT requires high energy expenditure and oxygen utilization when reabsorbing glucose in a hyperglycemic state. Renal tubular hypoxia occurs when oxygen consumption exceeds oxygen supply [15]. DKD is the result of persistent hypoxic conditions that induce ischemic injuries and interstitial fibrosis [21]. In response to hypoxia, various isoforms of hypoxia-inducible factors (HIF) regulate oxygen homeostasis [21]. Hyperglycemia is responsible for the unfavorable effects on HIF-1, such as the repression of HIF-1 signaling, destabilization of HIF-1, and inhibition of function [21]. In tubular oxygenation, an SGLT-2

inhibitor protects the kidneys by limiting the reabsorption activity, alleviating the oxygen demand, and averting the risk of progressing toward a hypoxic state [15].

Although sufficient oxygen supply is essential to avoid the risk of kidney injuries, in tubular energetics, the production of energy is paramount to maintaining kidney function and structural preservation. Hyperglycemia requires high energy-consumption for resorptive activity; subsequently, the ATP production shifts from favoring the fatty acid oxidation (FAO) pathway to glycolysis [15]. Through glycolysis, it is theorized that lipid accumulation can cause lipotoxicity by impacting the PCT's structure and function, leading to mitochondrial dysfunction [22]. SGLT-2i reshift the ATP-generation pathway back onto FAO, thus diminishing the risk of lipotoxicity and acute kidney injury (AKI) [15].

Hyperuricemia is associated with increased renal interstitial fibrosis [15]. SGLT-2i have uricosuric properties [19]. Urate can be released into the bloodstream through glucose transporter-9 (GLUT-9), which is expressed in the liver and PCT of the kidney [23]. However, glucose molecules have a higher affinity for GLUT-9 than urate molecules. When SGLT-2i enhance the glucose concentration in the renal tubule, glucose is then more likely to be transported across the renal epithelium into the bloodstream compared to urate. The untransferred urate is eliminated through the urine, alleviating the risk of interstitial fibrosis [15,23]. A recent systemic review found a decrease in gout flares in people taking SGLT-2i [24].

In people with diabetes, the expression of cell adhesion molecules, chemokines, and cytokines is increased in a manner that is pro-inflammatory [25]. The level of inflammation from chemokines and cytokines is low-grade, and its injury has far-reaching effects on the kidneys [25]. Interestingly, diabetes can increase the prevalence of M1 compared to M2 macrophages [25]. M1 are pro-inflammatory macrophages that inhibit cell proliferation and induce tissue damage, whereas M2 are anti-inflammatory macrophages that promote cell proliferation and tissue repair [25,26]. M1's effect can inhibit the degradation of the mesangial extracellular matrix, contribute to podocyte structural deformity, and increase glomerular macrophage infiltration through direct contact with the adhesion molecule ICAM-1 [25]. Although the mechanism for reduced inflammation with SGLT-2i remains elusive, SGLT-2i have been shown to be associated with reductions in the pro-inflammatory transcription of nuclear factor κappa B (NF-κB), interleukin-6 (IL-6), monocyte chemoattractant protein-1 (MCP-1), tumor necrosis factor receptor-1 (TNFR-1), matrix metalloproteinase-7 (MMP-7), and fibronectin-1 [15]. Hence, the kidney interstitium is protected from inflammatory insults and tubulointerstitial from fibrosis [15].

### 2.1.2. What Is the Evidence of SGLT-2 Inhibitors in Kidney Protection?

Nine randomized, double-blinded, multinational, placebo-controlled clinical trials investigated the effects of SGLT-2i on renal outcomes. Five of these trials (e.g., CANVAS Program, CREDENCE, DAPA-CKD, EMPA-REG OUTCOME, and EMPA-KIDNEY) conferred benefits in delaying the progression of renal impairment in patients with T2DM. An analysis of the secondary outcome focused on the renal effects in the VERTIS CV trial did not reveal clinical significance because the trial was not powered for secondary outcomes [27]. All of the trials' primary and secondary renal outcomes are summarized in Table 4. In Table 5, the renal outcomes for the landmark trials are summarized.

**Table 4.** Summary of Major Clinical Trials of SGLT-2 Inhibitors Available in the US Organized by Different Disease States.

| | Canagliflozin 100 mg, 300 mg | Dapagliflozin 5 mg, 10 mg | Empagliflozin 10 mg, 25 mg | Ertugliflozin 5 mg, 15 mg |
|---|---|---|---|---|
| Renal Trials | **CREDENCE (2019):** T2DM with albuminuric CKD, eGFR 30–89, UACR 300–5000, and were treated with RAS inhibitor: decrease kidney failure and CV events. No significant difference in rates of amputation or fracture. | **DAPA-CKD (2020):** eGFR 25–75 and UACR 200–5000: CKD patients with or without T2DM had a lower risk of in a sustained decline of: eGFR of at least 50%, ESKD, death from renal or CV causes, hospitalization from HF, and longer survival. | **EMPA-KIDNEY (2023):** Trial was stopped early due to clear positive efficacy in patients with CKD (eGFR 20–44 or eGFR 45–89 w/albuminuric). | **Analysis of VERTIS-CV (2020):** Decrease in the risk of sustained 40% decline in eGFR, less albuminuria, preserved eGFR over time in patients wtih T2DM and established ASCVD. |
| CV Outcomes Trials | **CANVAS (2017):** Decreased CV death, non-fatal MI or nonfatal stroke in T2DM patients at high risk of CV disease. However, increased amputation of the toe or metatarsal. | **DECLARE-TIMI (2019):** T2DM with or at risk for ASCVD: decreased CV death or hospitalization for HF. No difference in MACE (CV death, MI, or Stroke). | **EMPA-REG OUTCOME (2015):** Decreased MACE (CV death, non-fatal MI, non-fatal stroke) and of death from any cause vs. placebo among T2DM patients with high CV risk | **VERTIS-CV (2020):** Patients with T2DM and ASCVD, ertugliflozin was noninferior to placebo with respect to MACE. |
| Heart Failure Trials | **CHIEF-HF remote (2022):** Regardless of HF type and T2DM status 476 people HFrEF and HFpEF Study terminated early due to shifting priorities At 12 weeks, symptoms better with canagloflizin 100 mg [28] | **DAPA-HF (2019):** HF Class II-IV, LVEF ≤ 40%: Decreased risk of worsening HF (hospitalization or urgent care w/IV tx) or CV death, regardless if the patient was diabetic or not. **DELIVER (2022):** Primary endpoint met to reduce risk of CV death or worsening of HF in patients with HFpEF [29] | **EMPEROR-REDUCED (2020):** HF Class II-IV, LVEF ≤ 40%: Decreased CV death or hospitalization for HF, regardless if the patient was diabetic or not. **EMPEROR-PRESERVED (2020):** HF Class II-IV, LVEF >40%: Decreased CV death or hospitalization for HF, regardless if the patient was diabetic or not. | **Analysis of VERTIS-CV (2020):** T2DM with ASCVD: reduced first hospitalization for HF (HHF), total HHF, and total HHF/CV death. |

**Table 5.** Renal Outcomes.

| Randomized Control Trial (Sample Size) | Outcome Measure | Results | | | Explication (Of Statistical Significance) |
|---|---|---|---|---|---|
| | | **Treatment *** | **Placebo *** | **Hazard Ratio (95% CI);** ***p*-Value** | **Relative Risk, Absolute Risk, NNT, NNH** |
| | | **(Events/1000 Patient-Year)** | | | |
| CANVAS PROGRAM (*n* = 10,142) Canagliflozin 100 mg, 300 mg | Progression of Albuminuria (*n* = 9015 with normoalbuminuria or microalbuminuria at baseline) | 89.4 | 128.7 | 0.73 (0.67–0.79); *p* = 0.0184 ^ | RR = 27% |
| | Regression of Albuminuria | 293.4 | 187.5 | 1.70 (1.51–1.91); *p* = 0.4587 ^ | --- |
| | Renal Composite Outcome [a] | 5.5 | 9.0 | 0.60 (0.47–0.77); *p* = 0.3868 ^ | --- |
| CREDENCE (*n* = 4401) canagliflozin 100 mg | Primary Composite Outcome [b] | 43.2 | 61.2 | 0.70 (0.59–0.82); *p* = 0.00001 | RR = 30% |
| | End-Stage Kidney Disease [e] | 20.4 | 29.4 | 0.68 (0.54–0.86); *p* = 0.002 | RR = 31% |
| | Doubling of Serum Creatinine Level | 20.7 | 33.8 | 0.60 (0.48–0.76); *p* < 0.001 | RR = 40% |
| | Renal Death | 0.3 | 0.9 | N/A | --- |
| | Secondary Composite Outcome | 27.0 | 40.4 | 0.66 (0.53–0.81); *p* < 0.001 | RR = 34% |
| DAPA-CKD (*n* = 4304) dapagliflozin 10 mg | Primary Composite Outcome [d] | 4.6 * | 7.5 * | 0.61 (0.51–0.72); *p* < 0.001 | RR = 39% NNT = 19 |
| | Decline in eGFR of ≥50% | 2.6 * | 4.8 * | 0.53 (0.42–0.67); *p* < 0.05 | RR = 47% |
| | End-Stage Kidney Disease | 2.5 * | 3.8 * | 0.64 (0.50–0.82); *p* < 0.05 | RR = 36% |

**Table 5.** *Cont.*

| Randomized Control Trial (Sample Size) | Outcome Measure | Results | | | Explication (Of Statistical Significance) |
|---|---|---|---|---|---|
| | | Treatment * | Placebo * | Hazard Ratio (95% CI); *p*-Value | Relative Risk, Absolute Risk, NNT, NNH |
| | | (Events/1000 Patient-Year) | | | |
| EMPA-REG OUTCOME (Progression of Kidney Disease) (*n* = 7020) empagliflozin 10 mg, 25 mg | Incident or Worsening Nephropathy or Cardiovascular Death | 60.7 | 95.9 | 0.61 (0.55–0.69); *p* < 0.001 | RR = 39% |
| | Incident or Worsening Nephropathy | 47.8 | 76.0 | 0.61 (0.53–0.70); *p* < 0.001 | RR = 39% |
| | Progression to Macroalbuminuria | 41.8 | 64.9 | 0.62 (0.54–0.72); *p* < 0.001 | RR = 38% |
| | Doubling of Serum Creatinine Level Accompanied by eGFR of ≤45 mL/min/1.73 m$^2$ | 5.5 | 9.7 | 0.56 (0.39–0.79); *p* < 0.001 | RR = 44% |
| | Renal-Replacement Therapy | 1.0 | 2.1 | 0.45 (0.21–0.97); *p* = 0.04 | RR = 56% |
| | Doubling of Serum Creatinine Level Accompanied by eGFR of ≤45 mL/min/1.73 m$^2$, Initiation of Renal-Replacement Therapy, or Death from Renal Disease | 6.3 | 11.5 | 0.54 (0.40–0.75); *p* < 0.001 | RR = 46% |
| | Incident Albuminuria in Patients with a Normal Albumin Level at Baseline | 252.5 | 266.0 | 0.95 (0.87–1.04); *p* = 0.25 | --- |
| VERTIS-CV (*n* = 8246) ertugliflozin 5 mg, 15 mg | Renal Composite Outcome [c] | 0.9 * | 1.2 * | 0.81 (0.63–1.04); *p*-value not available | --- |

* = rate/events/participants per 100 patient-years; ˆ = *p*-value for homogeneity between CANVAS and CANVAS-R; [a] Composite Renal Outcome (CANVAS Program): Composite of sustained a 40% reduction in eGFR, need for renal replacement therapy, or death from renal causes; [b] Primary Composite Outcome (CREDENCE): doubling of serum creatinine level, end-stage kidney disease, renal death, or cardiovascular death; [c] Renal Composite Outcome (VERTIS CV): death from renal causes death, renal replacement therapy, or doubling of serum creatinine level; [d] Primary Composite Outcome (DAPA-CKD): reduction on eGFR at least 50%, end-stage kidney disease, death from renal causes, or death from cardiovascular causes; [e] End-Stage Kidney Disease (CREDENCE): dialysis initiated or kidney transplantation, or estimated GFR of <15 mL/min/1.73 m$^2$. CV, cardiovascular; MI, myocardial infarction; HHF, hospitalization for heart failure; RRT, renal replacement therapy.

### 2.1.3. Canagliflozin

The CANVAS Program consists of two trials: Canagliflozin Cardiovascular Assessment Study (CANVAS) and CANVAS-Renal (CANVAS-R). As the CANVAS-R Trial is more relevant to this section, the CANVAS Trial is discussed in the CV section. Over a mean follow-up period of 108 weeks, 5812 participants with T2DM at higher risk of CV events were monitored [30]. The prespecified outcomes include two renal composites: the first included a doubling of serum creatinine levels, ESKD, or death from renal causes; the second was similar to the first but used a 40% reduction in eGFR in place of serum creatinine levels [30]. The individual outcomes included a 40% reduction in eGFR, new onset of albuminuria at any level, and ESKD or death from renal causes [30]. The investigators of the CANVAS-R Trial concluded that canagliflozin showed renoprotective properties by reducing adverse kidney events in adults with T2DM, including worsening eGFR, albuminuria, and ESKD [30]. The renal outcomes included in CANVAS-R are summarized in Table 4.

The Canagliflozin and Renal Events in Diabetes with Established Nephropathy Clinical Evaluation (CREDENCE) Trial was designed to assess the effects of canagliflozin on participants with T2DM and kidney disease. This trial randomized 4401 participants in total and had a median follow-up duration of 2.62 years [31]. The CREDENCE Trial was terminated early due to the strong evidence of benefits observed from the primary composite outcome consisting of ESKD, a doubling of serum creatinine from baseline, and death from renal or CV causes [31]. The secondary outcomes were included for sequential hierarchical testing [31]. The investigators reported statistical significance associated with canagliflozin reducing the risk of kidney failure and CV events in people with T2DM and kidney disease [31]. All of the renal outcomes from CREDENCE are summarized in Table 4.

### 2.1.4. Dapagliflozin

The Dapagliflozin and Prevention of Adverse Outcomes in Chronic Kidney Disease (DAPA-CKD) Trial demonstrated that participants with CKD benefited from dapagliflozin treatment regardless of their T2DM status. Assessing 4094 participants with a 2.4-year median follow-up period, both the primary and secondary outcomes were met with statistical significance and are summarized in Table 4 [32]. The primary composite outcome was the first occurrence of a reduction in eGRF of 50% or greater, ESKD, and death from renal or CV causes [26]. The second renally focused outcome was similar to the primary outcome, except that it excludes death from CV causes [32]. Due to its clear renal-protective benefits, the DAPA-CKD Trial was discontinued prior to the projected end date [32].

### 2.1.5. Empagliflozin

The Empagliflozin Cardiovascular Outcome Event Trial in Type 2 Diabetes Mellitus Patients—Removing Excess Glucose (EMPA-REG OUTCOME) Trial was published in 2015, revealing its analysis of the CV outcomes. One year later, the investigators included a review to analyze empagliflozin's long-term effects on the kidneys [33]. The EMPA-REG OUTCOME Trial enrolled 7020 participants with T2DM and an elevated risk of CV events [34]. This trial consists of seven renal outcomes, including incidence or worsening nephropathy or CV death; incidence or worsening nephropathy; progression of macroalbuminuria; doubling of serum creatinine (SCr) level accompanied by eGFR $\leq$ 45 mL/min/1.73 m$^2$; renal replacement therapy (RRT); composite of double of SCr level accompanied by eGFR $\leq$ 45 mL/min/1.73 m$^2$, RRT, and incidence of albuminuria in patients with a baseline normal albumin level [34]. The analysis of the renal outcomes was reported with statistical significance, and all seven renal outcome measures are summarized in Table 4 [34]. This trial proved empagliflozin's linkage to kidney disease progression and lowering renal events in those with T2DM and elevated CV risk, regardless of impaired their kidney function status [34].

Additionally, empagliflozin in patients with chronic kidney disease (EMPA-KIDNEY) was the subject of a recent study, which observed a decrease in kidney disease progression

compared to the placebo, regardless of T2DM status [35]. The inclusion criteria included eGFR levels between 25 and 40 mL/min/1.73 m$^2$ or between 40 and 90 mL/min/1.73 m$^2$ with an albumin-to-creatinine ratio of at least 200 mg/g [35]. The primary renal composite outcome was time to first occurrence of ESKD, a sustained 40% decline in EGFR from baseline, and death from renal or CV causes [35]. In fact, this study was terminated earlier than anticipated due to the overwhelmingly positive efficacy in patients with CKD. Table 4 summarizes some of the essential results from this trial.

### 2.1.6. Ertugliflozin

The Evaluation of Ertugliflozin Efficacy and Safety Cardiovascular Outcomes (VERTIS CV) Trial conducted a secondary renal outcome among participants with T2DM and established ASCVD, which was not met with statistical significance [27]. The secondary outcome was defined as a renal composite that consisted of a doubling of serum creatinine level, renal replacement therapy (RRT), or death by renal causes [27]. The investigators conducted an analysis that was renally focused on the same sample of participants ($n = 8246$) from the VERTIS CV Trial, except the renal composite was redefined. The redefined renal composite included a sustained 40% reduction from baseline in eGFR instead of a measure in serum creatinine, while RRT and death by renal causes remained unchanged [27]. This secondary analysis was met with mixed results as the redefined renal composite and an individual component of the composite were statistically significant [27]. Table 4 lists the results from the initial and secondary analyses of the renal outcomes of the VERTIS CV Trial. The investigators concluded that ertugliflozin was associated with a reduced risk of kidney progression [27].

### 2.2. Clinical Implications

In 2013, the Food and Drug Administration (FDA) approved canagliflozin as the first SGLT-2 inhibitor for T2DM management [36,37]. Nearly a decade since the approval of canagliflozin, the newest additions to this class of agents available in the US include dapagliflozin, empagliflozin, ertugliflozin, bexagliflozin, and sotagliflozin. The evidence associated with SGLT-2i's effects on the kidneys has prompted the FDA to approve an expanded, kidney-related indication for both canagliflozin—used in type 2 diabetes—and dapagliflozin—with or without diabetes—for reducing end-stage kidney disease (Table 2). Although this class has a modest effect on glycemic management, the use of SGLT-2i has proven to be effective in the risk reduction of diabetes-related complications.

In 2019, the ADA first incorporated SGLT-2i into the clinical guidelines for renoprotective properties [5]. Since then, the ADA has continued to review new emerging data from trials—like EMPA-REG, CANVAS, CREDENCE, and DAPA-CKD—to continue its support of the use of SGLT2 inhibitors for renal protection. The CV and renoprotective benefits of SGLT-2i have recently designated this class of drugs as a first-line therapeutic option for adults with T2DM with or at high risk of ASCVD, HF, and/or CKD [5]. The ADA acknowledges that the trials have demonstrated a reduction in HF hospitalization, established CV disease, and CKD progression independent of glycemic management, which are all appropriate indications to initiate an SGLT-2 inhibitor [5].

Additionally, the Kidney Disease Improving Global Outcome (KDIGO) updated its guidelines for diabetes management in CKD in 2022. Patients initiated with an SGLT-2 inhibitor therapy must have a baseline eGFR $\geq$ 20 mL/min/1.73 m$^2$ [38]. The KDIGO guidelines are in agreement with ADA's recommendations in that an SGLT2 inhibitor with approved indication should be added for the purpose of reducing CKD progression, independent of glycemic management [38]. According to KDIGO's meta-analysis of EMPA-REG OUTCOME, CANVAS trials, CREDENCE, and DAPA-CKD, SGLT-2i are appropriate for patients with T2DM and eGFR $\geq$ 20 mL/min/1.73 m$^2$ [38]. The benefits recognized from these landmark trials have consistently demonstrated reductions in CV events and CKD progression.

The trials mentioned in this section have provided new insights into the application of SGLT2 inhibitors. The EMPA-REG Trial has shown that empagliflozin can delay kidney progression in patients with T2DM [33]. The CREDENCE Trial suggested that the renal benefits of SGLT2i are likely independent of glucose management; should the eGFR fall below 30 mL/min/1.73 m$^2$, canagliflozin can be an effective treatment to protect the kidneys [31]. The DAPA-CKD and EMPA-KIDNEY Trials generalized the applicability of SGLT-2 inhibitors to a broader population of individuals with CKD that does not center on T2DM status [32,33].

Understanding the beneficial effects of SGLT-2i on the kidneys and the barriers to prescribing them, what can be done? The recurring idea of advocacy has been a prevalent idea throughout the background research conducted for this review paper. Kim et al. created CKD-PCP, which is a model with two meanings to illustrate team-based care for T2DM [37]. The first meaning is geared toward medical providers: Cardiologists, Nephrologists, Diabetologists, and Primary Care Physicians, while the second meaning is related to the beneficial effects of SGLT-2i: CV disease, kidney disease, diabetes, and a reduction in blood pressure, calories, and plasma volume [37]. Schernthaner et al. wrote a manifesto for change and how to overcome clinical inertia [39]. Clinical inertia is defined as the aversion to change for reasons that are more associated with the physician [39]. The driving message is to utilize an SGLT2 inhibitor as long as an indication is met to mitigate the suboptimal outcomes of CKD, as explained in this section, by preserving the kidney functionality and structure.

Safety

With regards to the safety profile of SGLT-2i, the CREDENCE Trial, along with trials of other SGLT-2i, did not find an increased rate of amputation, as noted in the CANVAS Trial. After review, the Black Box Warning for leg and foot amputation listed for canagliflozin was removed by the FDA in 2020 [18]. SGLT-2i are safe and compatible with other oral glucose-lowering agents because this class of agents targets a novel pathway for glycemic management [37].

One of the most frequently reported adverse events of SGLT-2i are urinary tract infections (UTI) and genital infections [39]. The levels of bacteria and fungi in the genitourinary tract increase due to higher-than-normal concentrations of glucose in the urine [40]. These events are most frequent in the first few months of SGLT-2i use and are typically resolved through the standard treatment protocol [40].

SGLT-2i constrict the afferent arteriole of the glomerulus, which has a number of physiologic effects. In the early stage, the natureisis associated with SGLT-2i will lead to a transient reduction in the eGFR and potential rise in serum creatinine [41]. This can be present in the first 2 to 6 weeks but will return to baseline. This change in eGFR is expected and is still associated with improved renal outcomes in the long-term, similar to the RAAS agents in CKD.

Euglycemic diabetic ketoacidosis (DKA) is another risk to caution patients about, especially in those with insufficient beta cell activity (e.g., persons with type 1 diabetes), but such events are rare [42]. The proposed mechanism is thought to be based on SGLT-2i activating glucagon and counterregulatory hormones, leading to lipolysis and ketosis, and ultimately establishing an anion gap metabolic acidosis, which then triggers DKA [43]. Preoperative cessation of canagliflozin, dapagliflozin, and empagliflozin should start three days before surgery, while ertugliflozin should begin four days prior due to the elevated risk of DKA during this time period [44].

## 3. Cardiovascular Benefits

### 3.1. What Are the Proposed Theories or Potential Mechanisms of SGLT-2 Inhibitors in Providing CV Benefits?

SGLT-2i support blood pressure management without provoking hypotension [45]. By decreasing the intravascular volume, both the systolic and diastolic blood pressure experi-

ence a minor reduction [15]. In addition, they decrease renin release, thereby inhibiting the propagation of the signals to increase blood pressure.

SGLT-2i have been shown to provide CV benefits, specifically in the reduction in major adverse cardiovascular events (MACE) and CV mortality. Although there was a substantial reduction in CV deaths shown in the EMPA-REG Outcomes Trial, there was no significant reduction in stroke or myocardial infarction (MI) risk, suggesting that the reduction in CV deaths was not mediated through an atherothrombotic mechanism [46]. Instead, one of the proposed mechanisms linked to such benefits is the reduction in preload and afterload [46], which will be explained in depth in the next section regarding the provision of HF benefits.

In combination with inhibiting SGLT-2 cotransporters, SGLT-2i have recently been discovered to inhibit sodium-hydrogen exchange transporter 3 (NHE3), found in the proximal tubule [47]. Restoring a normal sodium concentration through natriuresis and osmotic diuresis thus improves the preload in heart ventricles [48]. A reduction in preload puts less strain on the heart muscles, thereby reducing CV outcomes. The proposed mechanism of achieving these CV benefits are through improving the cardiac metabolism, inhibiting sodium-hydrogen exchange transporters, reducing cardiac necrosis and fibrosis, as well as improving the production of adipokines, cytokines, and adipose tissue mass [48].

### 3.2. What Is the Evidence of SGLT-2 Inhibitors in Reducing CV Outcomes?

In 2008, the US FDA regulations mandated CV outcome trials (CVOTs) for glycemic-lowering agents to ensure CV safety [49]. The composite primary outcome in these CVOTs—EMPA-REG OUTCOME (empagliflozin), CANVAS Program (canagliflozin), DECLARE-TIMI 58 (dapagliflozin), and VERTIS-CV (ertugliflozin)—was the 3-point MACE, which consisted of CV deaths (including fatal stroke and fatal MI), non-fatal MI (excluding silent MI), and non-fatal stroke. Additionally, the CREDENCE Trial reported MACE data as a secondary outcome, which included CV deaths, MI, stroke, unstable angina, and hospitalization for HF [31]. The DAPA-HF (dapagliflozin) and EMPEROR-Reduced (empagliflozin) Trials did not use the 3-point MACE outcome but analyzed the effects of these SGLT-2i on the rates of CV deaths.

### 3.2.1. Canagliflozin

The CANVAS Program included the 3-point MACE outcome measure, comparing the incidence rates between the canagliflozin and placebo groups meeting this composite outcome. The data show that there was a 14% relative risk reduction (RRR) in the 3-point MACE occurrence in the canagliflozin group with statistical significance [50]. Interestingly, when the individual components of the 3-point MACE were examined, none of the RRRs met significance, suggesting none of the components were driving the composite outcome to be statistically significant [50]. The specific findings are summarized in Table 6.

The CANVAS Program's primary outcome was a composite of CV deaths, non-fatal MI, or non-fatal ischemic stroke, all commonly grouped as MACE; 65.6% of the enrollees had a history of CV disease [50]. The studies included 35.8% female and 78.3% White participants with an average age of $63.3 \pm 8.3$ years and an average duration of diabetes at $13.5 \pm 7.8$ years; the participants were monitored with a mean of 188.2 weeks [50].

The CREDENCE Trial included CV deaths as a component of the primary composite outcome and three secondary outcomes relating to the CV system (e.g., CV death, MI, or stroke), as summarized in Table 2. These secondary outcomes have shown significant reductions with canagliflozin, but a significant reduction was not achieved in CV deaths in the treatment group [31].

**Table 6.** Cardiovascular Outcomes.

| Cardiovascular Outcomes | | | | |
|---|---|---|---|---|
| **Randomized Control Trial** | **Outcome Measure** | **Results (Events/1000 Patient-Year)** | | **Hazard Ratio (95% CI); *p*-Value** |
| | | **Treatment *** | **Placebo *** | |
| CANVAS PROGRAM (*n* = 10,142) canagliflozin 100 mg, 300 mg | MACE [e] | 26.9 | 31.5 | 0.86 (0.75–0.97); *p* = 0.5980 ^ *p* = 0.02 for superiority; *p* < 0.001 noninferiority |
| | Death from CV Causes | 11.6 | 12.8 | 0.87 (0.72–1.06); *p* > 0.05 *p* = 0.9387 |
| | Non-fatal MI | 9.7 | 11.6 | 0.85 (0.69–1.05); *p* > 0.05 *p* = 0.9777 |
| | Non-fatal Stroke | 7.1 | 8.4 | 0.90 (0.71–1.15); *p* > 0.05 *p* = 0.4978 |
| EMPA-REG OUTCOME (*n* = 7020) empagliflozin 10 mg, 25 mg | MACE [e] | 37.4 | 43.9 | 0.86 (0.74–0.99); *p* = 0.04 for superiority; *p* < 0.001 noninferiority |
| | Death from CV Causes, Non-fatal MI, Non-fatal Stroke, or Hospitalization for Unstable Angina | 46.4 | 52.5 | 0.89 (0.78–1.01); *p* = 0.08 for superiority |
| | Death from CV Causes | 12.4 | 20.2 | 0.62 (0.49–0.77); *p* < 0.001 |
| | Non-fatal MI Excluding silent MI | 16.0 | 18.5 | 0.87 (0.70–1.09); *p* = 0.22 |
| | Non-fatal Stroke | 11.2 | 9.1 | 1.24 (0.92–1.67); *p* = 0.16 |
| DECLARE-TIMI 58 (*n* = 17,160) dapagliflozin 10 mg | MACE | 22.6 | 24.2 | 0.93 (0.84–1.03); *p* = 0.17 |
| | MI | 11.7 | 13.2 | 0.89 (0.77–1.01); *p* > 0.05 |
| | Ischemic Stroke | 6.9 | 6.8 | 1.01 (0.84–1.21); *p* > 0.05 |
| | Death from CV Cause | 7.0 | 7.1 | 0.98 (0.82–1.17); *p* > 0.05 |
| | CV Death or HHF | 12.2 | 14.7 | 0.83 (0.73–0.95); *p* = 0.005 |

**Table 6.** *Cont.*

| Randomized Control Trial | Outcome Measure | Results (Events/1000 Patient-Year) | | |
|---|---|---|---|---|
| | | Treatment * | Placebo * | Hazard Ratio (95% CI); *p*-Value |
| CREDENCE (*n* = 4401) canagliflozin 100 mg | CV Death | 19.0 | 24.4 | 0.78 (0.61–1.00); *p* = 0.05 |
| | CV Death or HHF | 31.5 | 45.4 | 0.69 (0.57–0.83); *p* < 0.001 |
| | CV Death, MI, or Stroke | 38.7 | 48.7 | 0.80 (0.67–0.95); *p* = 0.01 |
| | CV Death, MI, Stroke, or HHF or unstable angina | 49.4 | 66.9 | 0.74 (0.63–0.86); *p*-value not available |
| VERTIS CV (*n* = 8246) ertugliflozin 5 mg, 15 mg Note: events/100 patient-years | MACE [e] | 3.9 | 4.0 | 0.97 (0.85–1.11); *p* < 0.001 for non-inferiority |
| | CV Death | 1.8 | 1.9 | 0.88 (0.75–1.03); *p* > 0.05 |
| | Nonfatal MI | 1.7 | 1.6 | 1.04 (0.86–1.27); *p* > 0.05 |
| | Nonfatal Stroke | 0.8 | 0.8 | 1.00 (0.76–1.32); *p* > 0.05 |
| DAPA-HF (*n* = 4744) dapagliflozin 10 mg Note: events/100 patient-years | CV Death | 6.5 | 7.9 | 0.82 (0.69–0.98); *p* < 0.05 |
| | CV Death or HHF | 11.4 | 15.3 | 0.75 (0.65–0.85); *p* < 0.001 |
| EMPEROR-Reduced (*n* = 3730) empagliflozin 10 mg | Primary composite outcome [k] | 15.8 | 21.0 | 0.75 (0.65–0.86); *p* < 0.001 |
| | CV Death | 7.6 | 8.1 | 0.92 (0.75–1.12); *p* > 0.05 |
| EMPEROR-Preserved (*n* = 5988) empagliflozin 10 mg | Primary composite outcome [l] | 6.9 | 8.7 | 0.79 (0.69–0.90); *p* < 0.001 |
| | CV Death | 3.4 | 3.8 | 0.91 (0.76–1.09); *p* < 0.05 |

\* = rate/events/participants per 1000 patient-years; ^ = *p*-value for homogeneity between CANVAS and CANVAS-R; [e] MACE: Major Adverse Cardiovascular Events defined as CV death, non-fatal MI, or non-fatal ischemic stroke; [k] Primary Composite Outcome (EMPEROR-Reduced): hospitalization for heart failure, cardiovascular death; [l] Primary Composite Outcome (EMPEROR-Preserved): hospitalization for heart failure, cardiovascular death. CV, cardiovascular; MI, myocardial infarction; HHF, hospitalization for heart failure; RRT, renal replacement therapy.

### 3.2.2. Dapagliflozin

Unlike the trials mentioned above, the DECLARE-TIMI 58 Trial did not show a significant reduction in the incidence rates of 3-point MACE and its individual components in the treatment group [51]. The results are summarized in Table 6.

The Dapagliflozin Effect on Cardiovascular Events-Thrombolysis in Myocardial Infarction 58 (DECLARE-TIMI 58) Trial evaluated 17,160 participants who were blindly randomized to receive dapagliflozin 10 mg daily or a placebo [51]. The baseline demographics comprised a sample population with a mean age of 63.9 years and was dominant in males (63.1%) and White (79.7%) particpants. The mean eGFR was 85.2 mL/min/1.73 m$^2$, the median duration of diabetes was 11.0 years, 40.6% had an established CV disease, and 59.4% with multiple risk factors for CVD. For a median follow-up time of 4.2 years, DECLARE-TIMI 58's reported outcomes are summarized in Table 4 [51].

### 3.2.3. Empagliflozin

The EMPA-REG OUTCOME was the first CVOT to examine the safety of an SGLT-2 inhibitor specifically on the CV system. The baseline characteristics showed 3273 of the participants (46.6%) had a history of MI, and 1637 participants (23.3%) had a history of stroke [52]. The relative risk reduction (RRR) in the occurrence of MACE in the empagliflozin group was 14% with statistical significance, and a significant level was also achieved for the RRR in CV deaths, with 38% [52]. However, there were no significant differences between the placebo and empagliflozin with respect to the occurrence of non-fatal MI and stroke [52]. Some of the key findings from this trial are summarized in Table 6.

The EMPEROR-Reduced Trial investigated the effects of empagliflozin primarily in participants with HF. The study included the occurrence of CV deaths, and the difference between the treatment and the placebo groups was not statistically significant, as summarized in Table 6 [53].

### 3.2.4. Ertugliflozin

The VERTIS-CV Trial achieved its primary objective to demonstrate non-inferiority in the occurrence of the 3-point MACE compared to the placebo [27]. However, there was no significant difference in the rate of MACE or CV deaths compared to the placebo. Details of the results are summarized in Table 6.

### 3.2.5. Clinical Implications

The FDA has expanded the use of empagliflozin for reducing "the risk of cardiovascular death in adults with type 2 diabetes mellitus and established cardiovascular disease" [13] and canagliflozin for reducing "the risk of major adverse cardiovascular events in adults with type 2 diabetes mellitus and established cardiovascular disease" [12]. Independent of HbA1c levels and metformin therapy, canagliflozin or empagliflozin can now be used as a first-line treatment to "reduce the risk of major adverse CV events in adults with T2DM and established CVD" [5]. Although dapagliflozin did not have a significant reduction in the 3-point MACE in the DECLARE-TIMI 58 trial, both DECLARE-TIMI 58 and DAPA-HF have shown a significant reduction in CV death in the intervention group compared to the placebo [51,54].

In addition to glucose control, SGLT-2i may also be beneficial for cardiac risk factors, such as obesity or hypertension. In all of the CVOTs, the treatment group was associated with a decrease in body weight and systolic blood pressure [55,56]. Patients with T2DM on SGLT-2i lose approximately 2 to 3 kg [55]. SGLT2i have caused a significant reduction in blood pressure, as much as 1.66 to 6.9 mmHg in systolic BP and 0.88 to 3.5 mmHg diastolic BP [55,56].

In 2008, the FDA issued a new guidance draft to the pharmaceutical industry to assess CV risks during the development of new drugs for T2DM. As T2DM is associated with an increased risk of CV disease, the FDA wanted to assure prescribers and patients that the

antidiabetic medication in question is not associated with an unacceptable increase in CV risk [57].

Ertugliflozin showed non-inferiority but not superiority on MACE [27]. If an SGLT2i was enacted earlier, then it is estimated that as many as 920 deaths and 780 CV deaths, and 510 HHF could be avoided per 100,000 patients per year [39].

## 4. Heart Failure Benefits

### 4.1. What Are the Proposed Theories or Potential Mechanisms of SGLT-2 Inhibitors in Providing HF Benefits?

SGLT-2i have been shown to play a role in reducing the risk of hospitalization due to HF in patients with or without T2DM [53,54]. Although there have been no definitive mechanisms of action of SGLT-2i on the myocardium, the thought of indirect physiological and biochemical effects contribute to HF management. SGLT-2i were shown to have a combined effect on diuresis and vasculature. These agents promote osmotic diuresis, natriuresis, and glucosuria in patients with diabetes, thus leading to a reduction in extracellular fluid and blood pressure, as well as easing arterial stiffness [58]. Such a reduction then reduces cardiac preload and afterload [58], thus enhancing the systolic and diastolic functions. Various studies have demonstrated that SGLT-2i decrease blood pressure, arterial stiffness, and vascular resistance, which ultimately reduce afterload and improve myocardial oxygen supply [59–62]. All of this could contribute to the observed reduction in cardiac death and hospitalizations due to heart failure seen in some major SGLT-2i clinical trials.

### 4.2. What Is the Evidence of SGLT-2 Inhibitors in Reducing the Risks of Hospitalization due to HF?

A number of large-scale studies on SGLT-2i have been performed, providing evidence for the reduction in HF-related hospitalizations in patients with or without T2DM.

#### 4.2.1. Canagliflozin

Both the CANVAS Program and the CREDENCE Trial included hospitalization for HF as one of the outcomes, and the data from these trials showed a significant reduction in the number of events occurring in the canagliflozin group [30,31]. The results are listed in Table 7.

#### 4.2.2. Dapagliflozin

The DECLARE-TIMI 58 Trial, although not an HF-specific trial, showed that treatment with dapagliflozin resulted in a reduction in hospitalizations due to HF [51]. The incidence rates of hospitalization for HF were significantly lower within the dapagliflozin group compared to the placebo, as summarized in Table 7 [51].

The Dapagliflozin and Prevention of Adverse Outcomes in Heart Failure (DAPA-HF) Trial enrolled 4744 participants with or without T2DM, eGFR $\geq$ 30 mL/min/1.73 m$^2$, HF symptoms in the NYHA Class II, III, or IV, and a reduced left ventricular ejection fraction (LVEF) to 40% or less [54]. Each participant was blindly randomized to receive either dapagliflozin 10 mg or a placebo in a 1:1 ratio. The study had a median follow-up time of 18.2 months, a mean age of 66.2 $\pm$ 11.0 years, and the majority of participants were male (76.2%) or white (70.0%) [54]. Although this trial mainly focused on HF, a secondary renal composite outcome was included. In the DAPA-HF Trial there was no significant increase in adverse events occurring between the dapagliflozin arm and the placebo arm, HR 0.71; 95% CI (0.44–1.16) [54].

**Table 7.** Heart Failure Outcomes.

| Randomized Control Trial | Outcome Measure | Results (Events/1000 Patient-Year) | | Hazard Ratio (95% CI); *p*-Value |
|---|---|---|---|---|
| | | Treatment * | Placebo * | |
| CANVAS Program (*n* = 10,142) canagliflozin 100 mg, 300 mg | HHF | 5.5 | 8.7 | 0.67 (0.52–0.87); *p* < 0.05 *p* = 0.2359 |
| | CV mortality or HHF | 16.3 | 20.8 | 0.78 (0.67–0.91); *p* < 0.05 *p* = 0.4584 |
| EMPA-REG OUTCOME (*n* = 7020) empagliflozin 10 mg, 25 mg | HHF | 9.4 | 14.5 | 0.65 (0.50–0.85); *p* = 0.002 |
| | HHF or death from CV causes excluding fatal stroke | 19.7 | 30.1 | 0.66 (0.55–0.79); *p* < 0.001 |
| DECLARE-TIMI 58 (*n* = 17,160) dapagliflozin 10 mg | CV death or HHF | 12.2 | 14.7 | 0.83 (0.73–0.95); *p* < 0.005 |
| | HHF | 6.2 | 8.5 | 0.73 (0.61–0.88); *p* < 0.05 |
| CREDENCE (*n* = 4401) canagliflozin 100 mg | HHF | 15.7 | 25.3 | 0.61 (0.47–0.80); *p* < 0.001 |
| DAPA-HF (*n* = 4744) dapagliflozin 10 mg Note: events/100 patient-years | Primary composite outcome [j] | 11.6 | 15.6 | 0.74 (0.65–0.85); *p* < 0.001 |
| | HHF | 6.9 | 9.8 | 0.70 (0.59–0.83); *p* < 0.05 |
| EMPEROR-Reduced (*n* = 3730) empagliflozin 10 mg Note: events/100 patient-years | Primary composite outcome [k] | 15.8 | 21.0 | 0.75 (0.65–0.86); *p* < 0.001 |
| | HHF | 10.7 | 15.5 | 0.69 (0.59–0.81); *p* < 0.05 |
| EMPEROR-Preserved (*n* = 5988) empagliflozin 10 mg | Primary composite outcome [l] | 6.9 | 8.7 | 0.79 (0.69–0.90); *p* < 0.001 |
| | HHF | 4.3 | 6.0 | 0.71 (0.60–0.83); *p* < 0.05 |
| VERTIS-CV (*n* = 8246) ertugliflozin 5 mg, 15 mg Note: events/100 patient-years | HHF | 0.7 | 1.1 | 0.70 (0.54–0.90); *p* < 0.05 |

* = rate/events/participants per 1000 patient-years; [j] Primary Composite Outcome (DAPA-HF): hospitalization or an urgent visit for heart failure, hospitalization for heart failure, urgent heart-failure visit, cardiovascular death; [k] Primary Composite Outcome (EMPEROR-Reduced): hospitalization for heart failure, cardiovascular death; [l] Primary Composite Outcome (EMPEROR-Preserved): hospitalization for heart failure, cardiovascular death. CV, cardiovascular; MI, myocardial infarction; HHF, hospitalization for heart failure; RRT, renal replacement therapy.

### 4.2.3. Empagliflozin

In the EMPA-REG OUTCOME Trial, approximately 10% of the participants had HF at baseline [33]. The empagliflozin group had a 35% relative risk reduction (RRR) in hospitalization due to HF ($p < 0.002$) [47]. The EMPA-REG OUTCOME Trial included outcomes relating to HF, with a significant reduction in the number of events occurring in the empagliflozin group, as summarized in Table 7 [33].

The EMPEROR-REDUCED Trial—including primary composite renal outcome, e.g., chronic dialysis, renal transplant, or sustained 40% reduction in eGFR—enrolled 3730 participants with eGFR $\geq$ 20 mL/min/1.73 m$^2$ and established New York Heart Association (NYHA) Functional Class II-IV HF and a reduced left ventricular ejection fraction (LVEF) $\leq$ 40% for a median follow-up period of 1.33 years [53]. The prespecified composite renal outcome and its associated result are listed in Table 4. The participants had a mean age of 66.8 years old, and the majority of participants were white (70.4%) [53]. Similar to the EMPA-REG Trial, there was a significant reduction—when compared to the placebo—in the number of events that occurred in the empagliflozin group for the specified renal outcome.

### 4.2.4. Ertugliflozin

In the VERTIS-CV Trial, there was no statistical benefit for ertrugliflozin in the combined secondary outcome of death from any cause or hospitalization for heart failure (8.1% versus 9.1% HR 0.75–1.02, $p$ for superiority $p = 0.11$), but there was a significant reduction in the rates of HHF in the ertugliflozin compared to the placebo group (HR, 0.70; CI 0.54–0.90) [27]. These data are shown in Table 7.

### 4.2.5. Sotagliflozin

Although sotagliflozin is an inhibitor of both SGLT-1 and 2, it is mostly being considered for its SLGT-2-inhibition property. In March 2023, this medication was approved by the FDA to "reduce the risk of cardiovascular death, hospitalization for heart failure, and urgent heart failure visit in adults with heart failure or type 2 diabetes mellitus, chronic kidney disease, and other cardiovascular risk factors" [10]. The results from the SOLOIST-WHF Trial support this indication, with the rate of the primary endpoint—deaths from cardiovascular causes and hospitalizations and urgent visits for heart failure—occurrences being significantly lower in the sotagliflozin group (hazard ratio, 0.67; 95% CI, 0.52 to 0.85; $p < 0.001$) [63]. This trial randomized 1222 participants, with 79.1% (481/608 individuals) in the sotagliflozin group having a left ventricular ejection fraction below 50% vs. 79.0% (485/614 individuals) in the placebo group [63].

### 4.3. Clinical Implications

Although the renal and CV effects vary across the SGLT-2i class, all of the reviewed major SGLT-2i trials have shown a significant decrease in the risk of hospitalization for HF. Due to the consistent results across the class, there is no evidence to date of one SGLT-2i being superior to the other. The ADA, in its 2023 Standards of Care in Diabetes, recommended using an SGLT-2i for patients with type 2 diabetes who have HF or are at high risk of HF [5].

Notably, the DAPA-HF and EMPEROR-REDUCED Trials primarily focused on patients with HFrEF with or without type 2 diabetes. A similar risk reduction in hospitalization for HF was found between both groups, with or without diabetes. This suggests that SGLT-2i require more research in their MOA as CV benefits do not necessarily occur due to lowering serum glucose. Although most patients were already on a loop diuretic or mineralocorticoid receptor antagonist, they did not find that dapagliflozin was associated with a significant increase in adverse effects, such as volume depletion (7.5% vs. 6.8% $p = 0.40$) or worsening renal function (6.5% vs. 7.2% $p = 0.36$), compared to the placebo [51]. Empagliflozin was also not associated with a significant increase in volume depletion or worsening renal function [53].

Additionally, an improvement in glycemic control may not be the reason for empagliflozin's HF hospitalization reduction as the mortality curves diverged as early as one month [53]. However, a reduction in blood pressure, an alimoration of the volume overload status, and a correction of the tubuloglomerular feedback mechanism may play a key role in HF reduction and prevention [64]. As SGLT-2i exert their effects on the kidneys and are predominantly eliminated via the kidneys, the patient's renal function should be considered before starting an SGLT-2i or titrating the dose [5].

## 5. Public Health Impact

The reduction in mortality from chronic CV and kidney diseases suggests the impact of SGLT-2i on disease management and, recently, disease prevention [65,66]. The meta-analysis of the DELIVER, EMPEROR-Preserved, and SOLOIST studies found that SGLT-2i improved the overall population's health by decreasing symptoms and hospitalizations from HF [65]. Regarding T2DM prevention, dapagliflozin has been shown to reduce the incidence of diabetes in people with prediabetes [66].

Social issues within the healthcare system are the barriers for certain populations accessing novel therapies. It is worthwhile to discuss SGLT-2i in the context of social determinants of health (SDoH) as the lack of access to this class of medications creates health disparities/health inequities, thus exerting a public health impact, particularly in regard to many landmark trials showing cardiorenal benefits beyond these agents' glycemic effects. For example, SGLT-2i use is the lowest among the Black, Asian, and female demographics, whereas a median household income over six figures is associated with a higher rate of their usage [67]. This is especially relevant due to the rates of CV and renal complications being higher among individuals with lower socioeconomic status (SES) and who are more likely to struggle to gain access to healthcare [68].

The lack of pharmacoequity within the U.S. healthcare system can be attributed to a multitude of factors, including providers' comfort in prescribing newer therapies, high-cost sharing by patients, or providers' bias toward their patients with regard to health literacy and adherence to treatment [67]. However, even after removing the financial constraint variable, one Veterans Affairs (VA) study found Hispanic and Black individuals to have a considerably lower likelihood of being prescribed an SGLT-2 inhibitor than their White counterparts [69]. Additional efforts from the federal level are needed to address the lack of pharmacoequity in order to make this class of medications available to anyone who may need them for the management of type 2 diabetes and its comorbid conditions.

## 6. Discussion

SGLT-2i have continued to gain therapeutic indications and have shown important cardiorenal benefits. When the first SGLT-2 inhibitor was approved and used, the agent was perhaps viewed as another glycemic-lowering agent. As more evidence is being published about this class of agents, the clinical practice guidelines are constantly being revised to reflect their benefits in reducing MACE, preventing hospitalizations due to HF, and delaying the progression of renal decline in CKD.

According to the KDIGO 2022 guideline on diabetes in CKD, the first-line therapy for patients with type 2 diabetes and CKD is metformin and an SGLT-2 inhibitor; for using an SGLT-2 inhibitor, the eGFR should be at or above 20 mL/min/1.73 m$^2$ for initiating therapy, and the agent should be discontinued if the patient is started on dialysis [38]. On the other hand, a similar recommendation has been made in the ADA SOC 2023, suggesting those with CKD and type 2 diabetes—after maximizing an ACE inhibitor/ARB to a tolerable dose—an SGTL-2 inhibitor with primary evidence in delaying CKD progression should be considered as long as the eGFR is at or above 20 mL/min/1.73 m$^2$ [5]. In terms of type 2 diabetes and heart failure, with any symptoms of either HFrEF (heart failure with reduced ejection fraction) or HFpEF (heart failure with preserved ejection fraction), an SGLT-2 inhibitor with evidence of HF benefits should be initiated [5]. The last category, established ASCVD or indicators of high risk, is indicated to start an SGLT-2 inhibitor with

proven CVD benefits [5] [Established ASCVD includes myocardial infarction, stroke, or a revascularization procedure; indicators of high risk are age at or above 55, plus two or more of the following: obesity, hypertension, smoking, dyslipidemia, or albuminuria [5]].

Although the updated clinical guidelines support the use of SGLT-2i in managing diseases beyond type 2 diabetes, their real-world implementation is not without problems. Schernthaner et al. categorized five theories under clinical inertia, which is defined as the aversion to change for reasons that are more associated with the clinician [39]. Meanwhile, other theories may be financially motivated or unintentionally based on an individual's demographics [67]. For example, SGLT-2i were utilized more frequently if an individual had commercial insurance compared to Medicare Advantage, and the usage was higher in those with a higher median household income [67]. Medications that are cost-prohibitive may lead to prescription discontinuation and lead the patient to resort to medications that are cheaper and do not have the same level of evidence for CV and kidney benefits [38,68]. In another example, among the commercially insured population, there was an association of low prescribing rates in patients who were Black and female, many with HFrEF, ASCVD, CKD, and T2DM [69]. In 2018, 71.6% of endocrinologists were reported with the highest rates if prescribing SGLT-2i, second to metformin, when compared with 13.7% of cardiologists, 21.4% of family medicine physicians, and 23.3% of internal medicine physicians [70].

There seem to be some gender differences in how patients respond to this class of agents. A study conducted by Mirabelli and colleagues found a higher rate of discontinuing SGLT-2 inhibitors in females as a result of urinary tract infection [71]. In addition, Singh and Singh discovered that compared to males with diabetes, the cardiovascular benefits—captured in MACE (major cardiovascular event)—obtained from these agents are reduced in females [72]. More research is warranted to explore this identified difference.

*Another New Agent in the Class*

Bexagliflozin has been approved in 2023 for its glucose-lowering effect to be used in T2DM management in adults. The basic characteristics pertaining to this new agent are summarized in Table 8.

**Table 8.** Characteristics of Bexagliflozin [9].

| | |
|---|---|
| Indication | To improve glycemic management for T2DM in adults as an adjunct therapy adding to diet and exercise |
| Contraindication | • Hypersensitivity to bexagliflozin or its excipient<br>• Patients on dialysis |
| Warnings and Precautions | • Ketoacidosis<br>• Lower limb amputation<br>• Volume depletion<br>• Urosepsis and pyelonephritis<br>• Hypoglycemia<br>• Necrotizing fasciitis of the Perineum (Fournier's Gangrene)<br>• Genital mycotic infection |
| Common Adverse Reactions | • Female genital mycotic infections<br>• Urinary tract infection<br>• Increased urination |
| Impaired Renal Function | Not recommended in those with eGFR lower than 30 mL/min/1.73 m$^2$ |
| Note | Due to potentially increase the risk for diabetic ketoacidosis, not recommended in patients with T1DM |

The idea of advocacy and achieving health equity is a prevalent idea in researching the barriers to prescribing SGLT-2 inhibitors. With the influx of new data being published, continuous reviews on SGLT-2 inhibitors will help fill in the knowledge gaps and will be supported by the most recent evidence-based studies.

## 7. Conclusions

Although adverse events are an undeniable possibility associated with any medication, patient-specific factors should be determined when considering whether an SGLT-2 inhibitor is appropriate and if the benefits outweigh the risks. Due to the continuous positive evidence supporting the use of SGLT-2 inhibitors in those with type 2 diabetes, this class of medications has revolutionized the management of type 2 diabetes through their non-glycemic benefits. Notably, it has consistently been shown that the benefits of canagliflozin, dapagliflozin, and empagliflozin extend beyond glycemic management, significantly reducing hospitalization associated with heart failure and delaying the progression of kidney disease in those with type 2 diabetes.

As cardiovascular disease and kidney disease are the major sequelae and causes of death associated with this population, the non-glycemic benefits provided by SGLT-2 inhibitors indeed have profound implications for public health. Therefore, it would be a top priority to address the factors that drive health disparities or inequity for enhancing the number of patients receiving the newest guideline-driven pharmacotherapies, regardless of their backgrounds or demographics. Continued research efforts, public health policy advocacy, and clinical care initiatives must continue to be integrated to ensure that the usage of SGLT-2 inhibitors is full and equitable in the diverse population of individuals living with type 2 diabetes and its comorbidities.

**Author Contributions:** Conceptualization, C.F.Y. and J.H.S.; methodology, C.F.Y., N.F. and J.C.; writing—original draft preparation, C.F.Y., N.F., J.C. and J.H.S.; writing—review and editing, C.F.Y. and J.H.S. All authors have read and agreed to the published version of the manuscript.

**Funding:** This review paper received no external funding.

**Conflicts of Interest:** C.F.Y., N.F., and J.C. declare no conflict of interest. J.H.S. is/has been on the advisory boards for Abbott, AstraZeneca, Bayer, Eli Lilly, Nevro, and Novo Nordisk; he is/has been a consultant for Abbott, Bayer, and Novo Nordisk.

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
