# Peer review of "Exploring SGLT-2 Inhibitors: Benefits beyond the Glucose-Lowering Effect—What Is New in 2023?"

_endocrines, doi:10.3390/endocrines4030045_

Round 1

Reviewer 1 Report

This review explores the SGLT2i class of anti-diabetic drugs, with particular reference to cardiovascular and renal benefits. The main strength is that this is a prominent topic in the treatment of diabetes and beyond. The weakness is that there are many good reviews on this issue and the authors should try to add novelty and a personal interpretation to their manuscript, for example by foreseeing future directions. I would recommend to improve the following major points:

1. The overview on SGLT-2i (page 3) is scant and could be improved, with the addition of the main metabolic (glycemic and ponderal) effects discussed. Table 1 is not even cited in the text and the source of the data reported should be supported by the relative bibliography.

2. Fig. 1 anticipates unexplained concepts, should be more detailed and attractive to the readers (with colours?) and could be presented at the end of the description of renal and cardiovascular benefits (page 25). Currently known liver effects of SGLT2i could also been mentioned in the figure, even if data on NAFLD are preliminary. Mechanisms leading to renal and CV protection could be added as well (or briefly commented in the figure legend).

3. Adverse effects (page 5). Many effects should be accompanied by pertinent bibliography. Same observation in the first paragraph Heart Failure Benefits at page 22.

4. Table 8, mentioned at page 27, has been omitted in the manuscript. Please provide it in the revised version. 

5. A paragraph considering “Future directions” may added at the end of the paper, before the conclusion, to include novel ideas and personal interpretations, in agreement with what expected from the title. 

6. Also, a note on gender differences could be useful. For example, Mirabelli found a higher discontinuation of these drugs in females due to UTI; Rivera found reduced cardiovascular benefits in females.

6. In general, headings and subheadings could be improved, as they are not always clear.

7. Bibliography for such a wide and important topic could be expanded. Among the suggested pertinent papers:

Zelniker TA, et al. Lancet 2019

Mirabelli M, et al. J Diabetes Res 2019

Cowie MR & Fisher M. Nature Rev Cardiol 2020

Xu B, et al. Cardiovasc Diabetol 2022

Rivera FB, et al. Am Heart J Plus Card Res Pract 2023.

Minor issues:

- Overall, there is an overuse of abbreviations, even without further citations, as for DSMES at page 3. Also, for unusual terms, this does not help the reader. I would omit SOC for Standard of Care. Please revise.

- There are some typo errors to be fixed, even starting from the frontpage. 

- In the abstract, I would specify that “access to these agents has not been equal, at least in the USA” or similar locution.

- page 6, last line: a person without diabetes…. This is confusing, as the threshold for glucose is about 180 mg/dL for all. Consider rephrase as following “Renal threshold for glucose is..., which normally allows glucose to be reabsorbed…”

- page 27: I would highlight that Bexaglifozin has been approved in 2023 (to be in line with the title of the review…)

I recommend to format the paper according to the journal template.  

Author Response

Reviewer 1

This review explores the SGLT2i class of anti-diabetic drugs, with particular reference to cardiovascular and renal benefits. The main strength is that this is a prominent topic in the treatment of diabetes and beyond. The weakness is that there are many good reviews on this issue and the authors should try to add novelty and a personal interpretation to their manuscript, for example by foreseeing future directions. I would recommend to improve the following major points:

  1. The overview on SGLT-2i (page 3) is scant and could be improved, with the addition of the main metabolic (glycemic and ponderal) effects discussed. Table 1 is not even cited in the text and the source of the data reported should be supported by the relative bibliography.

Response to the comment: Thanks for pointing this out. We have fixed Table 1 according to the comment. However, the first paragraph under the heading “An Overview of SGLT-2 Inhibitors” is meant to be brief, with details coming after.

  1. Fig. 1 anticipates unexplained concepts, should be more detailed and attractive to the readers (with colours?) and could be presented at the end of the description of renal and cardiovascular benefits (page 25). Currently known liver effects of SGLT2i could also been mentioned in the figure, even if data on NAFLD are preliminary. Mechanisms leading to renal and CV protection could be added as well (or briefly commented in the figure legend).

Response to the comment: Thanks for the suggestions. After much consideration, we have decided to cut out this section, including Figure 1, as we don’t think this section adds much value.

  1. Adverse effects (page 5). Many effects should be accompanied by pertinent bibliography. Same observation in the first paragraph Heart Failure Benefits at page 22.

Response to the comment: Thank you for catching these. Appropriate citations have been added.

  1. Table 8, mentioned at page 27, has been omitted in the manuscript. Please provide it in the revised version. 

Response to the comment: Thank you very much for catching this. The statement has been removed, as we feel Table 8 is no longer needed with the information and substance included in this paragraph.

  1. A paragraph considering “Future directions” may added at the end of the paper, before the conclusion, to include novel ideas and personal interpretations, in agreement with what expected from the title. 

Response to the comment: While we see the point of the comments, the idea of “future directions” has been exemplified in the latter portion of the paper in the “Public Health Impact” section and the “Discussion” section.

  1. Also, a note on gender differences could be useful. For example, Mirabelli found a higher discontinuation of these drugs in females due to UTI; Rivera found reduced cardiovascular benefits in females.

Response to the comment: Thank you for the suggestion. We have incorporated the suggestion on Page 27.

  1. In general, headings and subheadings could be improved, as they are not always clear.

Response to the comment: We have fixed the formatting of the manuscript, but we are not sure if this helps address this comment. We would appreciate more specific guidance.

  1. Bibliography for such a wide and important topic could be expanded. Among the suggested pertinent papers:

Zelniker TA, et al. Lancet 2019

Mirabelli M, et al. J Diabetes Res 2019

Cowie MR & Fisher M. Nature Rev Cardiol 2020

Xu B, et al. Cardiovasc Diabetol 2022

Rivera FB, et al. Am Heart J Plus Card Res Pract 2023.

Response to the comment: While we appreciate the list of references provided above, we are not quite sure how to address this comment or suggestion. Please provide more specific guidance as to which section or sections of this manuscript would need expansion. Thank you!

Minor issues:

- Overall, there is an overuse of abbreviations, even without further citations, as for DSMES at page 3. Also, for unusual terms, this does not help the reader. I would omit SOC for Standard of Care. Please revise.

Response to the comment: While we respect the comment, we are confused as to why using abbreviations would be a problem. Every time an abbreviation is used, we always spell out the full term, e.g., diabetes self-management education and support (DSMES) on page 2 and Standards of Care in Diabetes (SOC) on page 2. We will need more guidance on what terms – considered as “unusual” terms -- should or should not be abbreviated. Please advise.

- There are some typo errors to be fixed, even starting from the frontpage. 

Response to the comment: Thank you for pointing this out. We have read through the draft again and caught the errors as we were able to.

- In the abstract, I would specify that “access to these agents has not been equal, at least in the USA” or similar locution.

Response to the comment: Thank you for the suggestions. We have incorporated the comment.

- page 6, last line: a person without diabetes…. This is confusing, as the threshold for glucose is about 180 mg/dL for all. Consider rephrase as following “Renal threshold for glucose is..., which normally allows glucose to be reabsorbed…”

Response to the comment: Thank you for pointing this out. The comment has been incorporated.

- page 27: I would highlight that bexaglifozin has been approved in 2023 (to be in line with the title of the review…)

Response to the comment: Thank you for the suggestion. It was mentioned on Page 3, and we also added the suggestion on Page 27.

I recommend to format the paper according to the journal template.  

Response to the comment: Thank you for the suggestion. The paper has been slightly formatted to meet the manuscript submission requirements under Review.

Reviewer 2 Report

This is a review article regarding efficacy and safety of SGLT2 inhibitors. The manuscript is well-written. There are a few minor comments.

1.        Product names may be removed from the tables.

2.        Page 24. Box 1 may be removed.

3.        It is better if mechanisms of SGLT2 inhibitors for cardiorenal protection are more precisely described and discussed.

4.        It is better if the authors could discuss the differences among the SGLT2 inhibitors.

Author Response

Reviewer 2

This is a review article regarding efficacy and safety of SGLT2 inhibitors. The manuscript is well-written. There are a few minor comments.

  1. Product names may be removed from the tables.

Response to the comment: Thank you for the suggestion. We think it would be important to have the brand names at least be included in one table, Table 1. We have removed brand names from all other tables.

  1. Page 24. Box 1 may be removed.

Response to the comment: Thank you for the suggestion. However, we have a difference in opinion, as bexagliflozin is the newest SGLT-2 inhibitor, with no detailed information included in the early portion of the review paper. It would be important for the readers to know this information; for example, not every SGLT-2 inhibitor has the same renal cutoff.

  1. It is better if mechanisms of SGLT2 inhibitors for cardiorenal protection are more precisely described and discussed.

Response to the comment: Thanks for the recommendation. However, we can better respond to this suggestion if this recommendation were more specific.

  1. It is better if the authors could discuss the differences among the SGLT2 inhibitors.

Response to the comment: We were wondering what specifics this comment referred to, as Tables 1 to 4 listed the differences among the four SGLT-2 inhibitors.

Round 2

Reviewer 1 Report

The authors have sufficiently addressed most of the comments raised. As suggested, a coloured figure recapitulating the main mechanisms underpinning extra-metabolic SGLT-2i effects could have been attractive to the readers, but is not indispensable.